# Impact of Steaming Mode on Chemical Characteristics and Colour of Birch Wood

**Anton Geffert \*, Jarmila Geffertová, Eva Výbohová and Michal Dudiak**

Faculty of Wood Sciences and Technology, Technical University in Zvolen, T.G.Masaryka 24, 96001 Zvolen, Slovakia; geffertova@tuzvo.sk (J.G.); vybohova@tuzvo.sk (E.V.); xdudiak@is.tuzvo.sk (M.D.)
\* Correspondence: geffert@tuzvo.sk; Tel.: +421-45-520-6378

**Abstract:** The aim of this work was to evaluate the changes of the chemical components in birch wood (*Betula pendula* Roth) caused by steaming with saturated steam at three temperatures—105 °C, 125 °C and 135 °C. In the samples of the original wood and wood after steaming, select chemical characteristics were determined, and wood, isolated holocellulose and Seiferts's cellulose were analysed by attenuated total reflectance Fourier transform infrared (ATR-FTIR) spectroscopy. The greatest changes in the birch wood characteristics were observed in steaming mode III (135 °C). The differential spectra of the birch wood samples indicated that the hemicelluloses were significantly degraded and that the dehydration reactions were able to proceed. A clear increase in both unconjugated and conjugated carbonyls was seen only in mode III. The findings also confirmed the greater sensitivity of the guaiacyl lignin contained in broadleaves to elevated steaming temperatures, as well as the course of the thermal oxidation reactions and the formation of new carboxyls in mode III. The decrease in the ratio of absorbances H $_{1732/2900}$ and H $_{1243/2900}$ demonstrated the cleavage of acyl (acetyl and formyl) groups from birch wood hemicelluloses. The qualitative and quantitative changes of the hemicelluloses and extractive substances in birch wood during steaming were well-correlated with the measured pH values and wood colour.

**Keywords:** birch wood; saturated steam; extractives; polysaccharides; cellulose; lignin; pH value; ATR-FTIR spectroscopy; CIEL\*a\*b\*

## 1. Introduction

Hydrothermal treatment is a method often applied to pretreat wood materials, which must obtain certain external and internal wood properties, for example, relating to colour, plasticity and durability. During hydrothermal treatment, the wood is treated with water respectively with water vapor at different modes—temperature, pressure, time, hydro module [1]. Moisture presence in the wood is a necessary condition for the chemical reaction runs and to achieve the homogeneous discolouration throughout the whole cross section of wood. Chen et al. [2] used saturated steam at a lower temperature and at a higher humidity of wood for obvious discolouration with no effect on polysaccharide components. Although the saturated steam method effectively induces discolouring, it has so far received less attention by the wood industry. Goal-directed changes in colour by hydrothermal treatment of full volumes of wood have great practical importance, especially in the production of furniture, decorative products and other wood products [3].

The basic chemical processes taking place in the hydrothermal treatment of wood include catalytic hydrolytic processes, which concern the hydrolytic reactions of polysaccharides, lignin–saccharide bond disruption and the hydrolysis of lignin. Hydrothermal action rapidly dissolves hemicellulose and acts slightly slower on the water-soluble portion of lignin and (to a lesser extent) also affects the amorphous portion of cellulose. Water-soluble accessory ingredients are extracted from wood during

hydrothermal action; these can include inorganic salts, monosaccharides and oligosaccharides, various polysaccharides (e.g., starch and pectins), cyclic alcohols, dyes, tannins and some low molecular weight phenols [4,5].

At low hydrothermal temperatures (60–80 °C), there are negligible chemical and structural changes in the basic wood components. Increasing the temperature of the hydrothermal reaction from 100 °C to 150 °C deepens the chemical and physicochemical changes of all components of the wood substance [6].

The hydrothermal treatment process is accompanied by changes in the colour of the processed material, the intensity of which is determined by the treatment conditions and the wood species. Reactions of the wood substance degradation products (e.g., condensation and oxidation of 2-furaldehyde, reactions of carbohydrates and pectins), as well as chemical changes in the extractives and lignin, influence the altered wood colouration. The mechanism of colour change is complex and involves a number of overlapping reactions of the basic wood components and their degradation products [7,8].

Wood colour changes are believed to occur as a result of intramolecular dehydration reactions and are associated with a conjugated system of multiple bonds and the free electron pairs of oxygen in the phenolic hydroxyl group. Increasing the content of phenolic hydroxyl groups in lignin affects its spectral characteristics and may result in the formation of other chromophores, such as carbonyl and carboxyl groups [6,9].

An increased intensity of the degradation reactions in the wood is achieved by increasing the temperature and the concentration of hydroxide cations. The gradual increase in acidity of the environment is caused by cleavage of the acetyl and formyl groups from hemicelluloses. The often observed lowering of pH below the values corresponding to the acyl group content in the wood is due to secondary reactions of the degradation products of the carbohydrate moiety.

It is difficult to explain the mechanism of the combined effect of heat and water on the properties of lignin in wood, as it is a complex, multifactor process in which changes in the polysaccharide fraction also play an important role. At temperatures of 80 °C to 100 °C, the simultaneous course of competitive depolymerisation and condensation reactions of lignin have been demonstrated. Assessing the effect of lignin condensation in the later stages of hydrothermal action is complicated by the fact that at temperatures of 80 °C to 140 °C, the degradation of the hemicellulose fraction also occurs while reducing the strength of the wood. The amount of hemicellulose degradation products passing into hydrolysates and condensates during heating and steaming is approximately twice that of the amount of lignin, suggesting a higher rate of degradation [6].

The most abundant component of wood, cellulose, is the most stable under hydrothermal conditions. The cellulose content of the wood increases relative to the duration of the hydrothermal treatment in the temperature range of 80 °C to 140 °C and is in proportion to the decrease in the hemicellulose and lignin components. At temperatures above 100 °C, partial depolymerisation of the cellulose begins [6].

The aim of this work was to evaluate changes in select chemical characteristics of birch wood caused by steaming with saturated steam under operating conditions for 12 h at three different temperatures—105 °C, 125 °C and 135 °C—and to identify and explain the changes of select chemical characteristics that occur upon changes in the pH value and wood colour using ATR-FTIR spectroscopy.

## 2. Materials and Methods

### 2.1. Material

The samples of birch wood (Betula pendula Roth) supplied from an industrial plant Sundermann Ltd. (Banská Štiavnica, Slovakia) were used to investigate chemical changes that occurred in different steaming treatments. Birch wood samples with dimensions of 32 mm × 90 mm × 600 mm, density $\rho = 704 \pm 61$ kgm$^{-3}$ and humidity $w_a > 45\%$ were thermally treated with saturated steam in an APDZ

240 pressure autoclave (Himmasch AD, Haskovo, Bulgaria) [10,11]. The steaming temperature in each hydrothermal treatment mode is shown in Table 1.

**Table 1.** The parameters of hydrothermal treatment of birch wood.

| Steaming Mode | Temperature of Saturated Water Steam (°C) | Duration (h) |
|:---:|:---:|:---:|
| I | 105 ± 2.5 | |
| II | 125 ± 2.5 | 12 |
| III | 135 ± 2.5 | |

## 2.2. Methods

To monitor the chemical changes, disintegrated samples of the original birch wood and the wood after steaming were used (0.5–1.0 mm fraction of sawdust prepared from completely disintegrated boards including surface and centre part).

In the samples of the original wood and the wood after steaming, the following chemical characteristics were determined:

**Ethanol–toluene solubility of wood:** procedure according to ASTM D 1107-96: 2 g of oven-dry sawdust were extracted with ethanol-toluene (2:1) for 7 h in a Soxhlet apparatus. The resulting extract was distilled off in a vacuum evaporator, dried in an oven at $t = 105$ °C to constant weight and the amount of extractive substances was determined gravimetrically [12].

**Polysaccharide fraction (holocellulose):** method according to Wise: repeated (3-fold) treatment of $NaClO_2$ (1.5 g) in the presence of acetic acid (10 drops) on extracted sawdust (5 g) for 1 h at $t = 80$–90 °C. After washing (water, acetone), cooling and filtering, the sample was dried in an oven at $t = 105$ °C to constant weight and the amount of holocellulose was determined gravimetrically [13].

**Cellulose:** Kurschner–Hoffer method: repeated (3-fold) boiling of 1 g of extracted sawdust with a mixture of concentrated $HNO_3$ and 95% ethylalcohol (1:4) in a flask under reflux for 1 h. After filtering, washing (ethanol + $HNO_3$, hot water) and drying in an oven at $t = 105$ °C to constant weight, the amount of cellulose was determined gravimetrically [13].

**Lignin:** procedure according to ASTM D 1106-96: two-stage treatment with sulfuric acid (step 1: 1 g of extracted sawdust was treated with 15 $cm^3$ of 72% $H_2SO_4$, intense stirring for 1 min at $t = 12$–15 °C, followed by standing for 2 h at room temperature (20 °C); step 2: transfer quantitatively to a boiling flask, dilute to 3% concentration with 560 $cm^3$ of distilled water and reflux for 4 h). The brown lignin precipitate settled, filtered through a weighed glass filter and thoroughly washed with hot water (500 $cm^3$) followed by drying in an oven at $t = 105$ °C to constant weight and the amount of lignin was determined gravimetrically [14].

**Seifert's cellulose:** acetylacetone method: 1 g of extracted sawdust was heated in a boiling water bath for 30 min in a mixture of acetylacetone (2 $cm^3$), dioxane (1 $cm^3$) and concentrated HCl (0.75 $cm^3$). Cooling was followed by addition of 20 $cm^3$ of methanol, filtering through weighed glass frit (S3), washing with methanol (50 $cm^3$), hot water (20 $cm^3$), dioxane (20 $cm^3$) and methanol (25 $cm^3$). Subsequently, the sample was dried to constant weight in an oven at $t = 105$ °C and the amount of Seifert cellulose was determined gravimetrically [15].

The pH measurement of the original birch wood and the wood after steaming was performed by the direct method using a LanceFET+H pH meter (SI 600) with a piercing probe. Three measurements of each sample were carried out in the centre and 100 mm away from the edge. The sample was drilled using an electric drill machine, and the resulting sawdust was pressed back into the opening; subsequently, a piercing electrode was applied. After achieving appropriate contact between the sawdust and probe, changes in the pH value were recorded every 15 s for a 7-min period [16].

The original and thermally treated birch wood samples and the isolated holocellulose and Seiferts's cellulose from these samples were analysed by ATR-FTIR spectroscopy. The measurements were carried out using a Nicolet iS10 FTIR spectrometer equipped with Smart iTR attenuated total reflectance (ATR) (a sampling accessory with diamond crystal—Thermo Fisher Scientific, Madison WI, USA).

The resolution was set at 4 cm$^{-1}$, 32 scans were recorded for each analysis, and the wavenumber range was from 4000 cm$^{-1}$ to 650 cm$^{-1}$. Six analyses were performed per sample. The spectra were evaluated using the OMNIC 8.0 software (Thermo Fisher Scientific, Madison WI, USA).

Colour measurement was performed on the treated original birch wood and on the wood after steaming using the Color Reader CR-10 (Konica Minolta, Japan) with an illuminated area of 8 mm. All colour measurements were taken using conditions of the standard illuminant D65 and Specular Component Included (SCI). Six measurements were performed on each sample, where the lightness L* and the colour coordinates a* and b* were evaluated. A total of 3 measurements were performed at the centre and 100 mm from the edge along the height of the block, and 3 measurements were performed at the centre and 100 mm from the edge along the width of the block. Measurements were performed on 6 samples in each mode. From the difference of the colour coordinates (ΔL*, Δa* and Δb*), the total colour difference ΔE* was determined according to the following equation [17]:

$$\Delta E* = \sqrt{\left(L_2^*-L_1^*\right)^2+\left(a_2^*-a_1^*\right)^2+\left(b_2^*-b_1^*\right)^2} \tag{1}$$

where (L$_2$*−L$_1$*) change in value of black-white coordinate (specific lightness)

(a$_2$*−a$_1$*) change in value of green-red coordinate

(b$_2$*−b$_1$*) change in the value of blue-yellow coordinate

## 3. Results and Discussion

The chemical analysis results of the original birch wood samples and wood samples after steaming in each mode are shown in Figure 1.

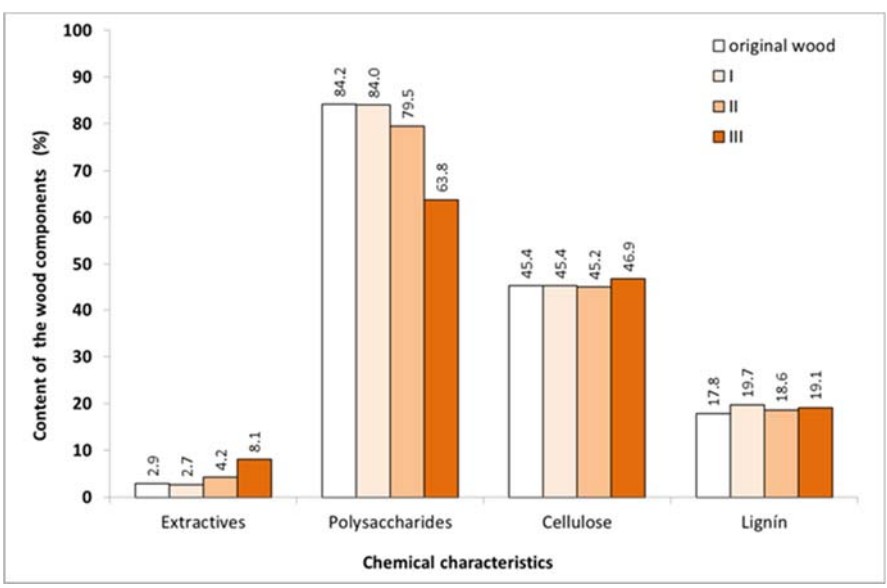

**Figure 1.** Chemical characteristics of original birch wood and after steaming.

The greatest changes in the birch wood after steaming were observed in the polysaccharides, with the greatest decrease in polysaccharide content resulting from mode III—a steam temperature of 135 °C. At this temperature, a small relative increase in the cellulose content caused by the degradation of the more labile wood components was apparent.

In the observed steaming regimes, the lignin content increased slightly compared to the original birch wood sample, and this increase was also attributable to the degradation of the more labile wood components.

The changes recorded in the content of extractive substances can be attributed to a large number of parallel reactions, in addition to the degradation of existing substances and new degradation

products being formed. The overall increase in the content of extractives was predominantly due to the degradation and destruction of the hemicellulose portion of the polysaccharide fraction in the wood.

The depth of change in the chemical composition of wood depends on the hydrothermal treatment conditions. The main role in hydrothermal action is played by nascent organic acids (acetic acid and formic acid), which are formed by cleavage of the acetyl and formyl groups of hemicelluloses. These volatile organic acids cause the destruction of hemicelluloses and, in part, contribute to the amorphous content of cellulose, as well as result in the dissolution of lignin and, in later phases, degrade monosaccharides. Organic acids released from wood catalyse various hydrolytic, dehydration and degradation reactions of carbohydrates and their products, but they also participate in condensation reactions [5].

Figure 2 details the changes in the polysaccharide fraction in birch wood according to the steaming mode. The proportion of polysaccharides decreased with increasing steaming temperature, while the degradation of polysaccharides accelerated with increasing steaming temperature. A decrease of 0.2% was observed at 105 °C (mode I), 4.7% at 125 °C (mode II) and 20.4% at 135 °C (mode III). Since the proportion of cellulose in the steaming process changed only minimally, nearly the entire decrease can be attributed to the loss of hemicellulose (Figure 3). The results obtained are in agreement with the conclusions of Kačík [18], according to which the loss of holocellulose in hydrothermally treated wood occurs mainly through the degradation of non-cellulosic polysaccharides.

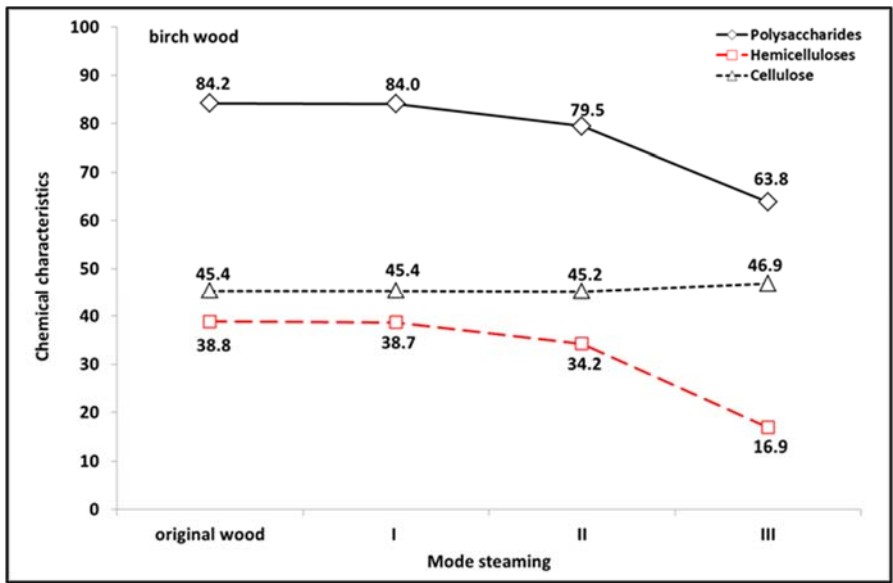

**Figure 2.** Changes in polysaccharide fraction of birch wood according to steaming mode.

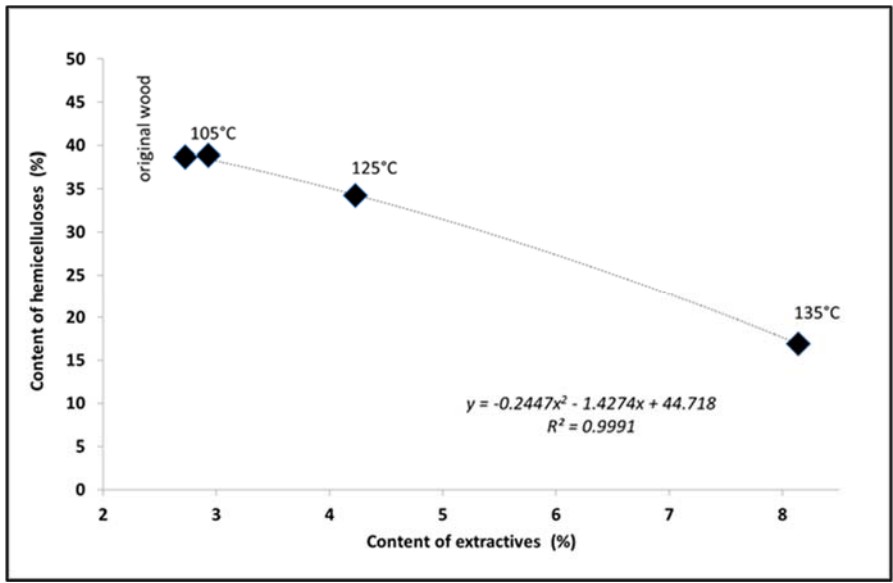

**Figure 3.** Dependence of the contents of hemicelluloses and extractives.

The process of releasing acidic components from wood and the associated change in pH in individual steaming modes is illustrated in Figure 4. The weakly acidic pH 5.3 of the original wood shifted toward the acidic region with increasing steaming temperature due to the more intense formation of acidic components (especially acetic acid and formic acid), with the pH decreasing to 3.2 in mode III.

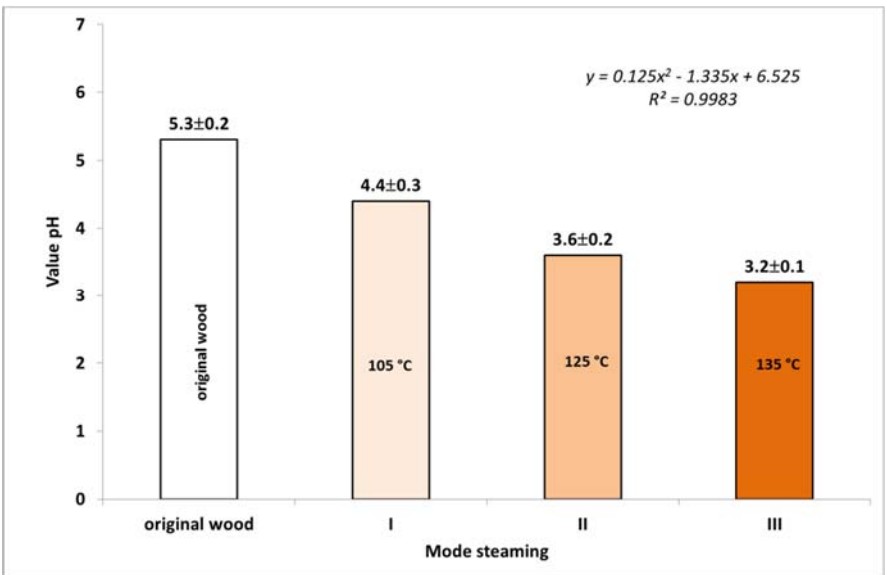

**Figure 4.** Change in the pH value according to steaming mode.

The changes in the contents of hemicelluloses and extractives as a function of pH showed that a significant shift of pH to the acidic region below 4.0 was accompanied by greater degradation of hemicelluloses and, consequently, an increased production of extractable substances (Figure 5).

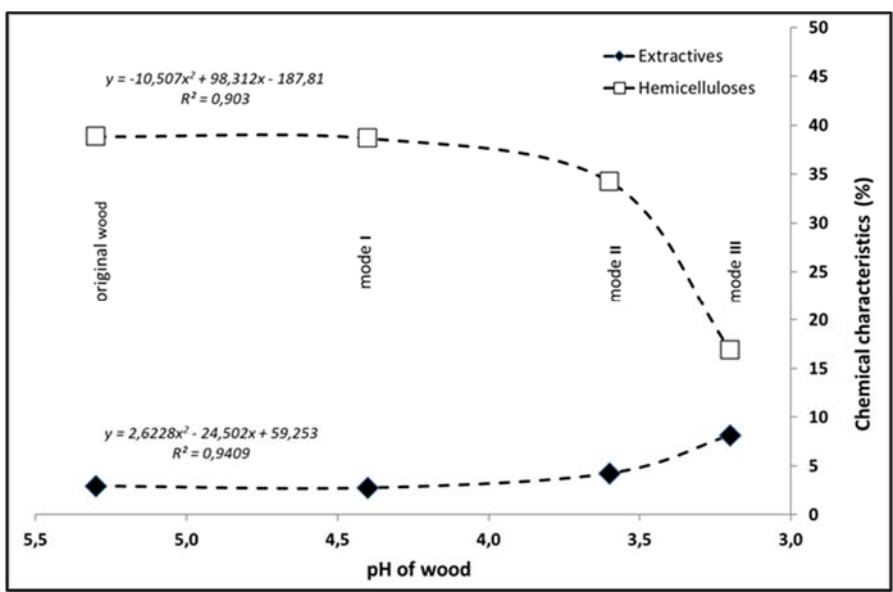

**Figure 5.** Dependence of select chemical characteristics on pH.

The differences in ATR-FTIR absorption spectra were used to further elucidate the changes in chemical substances taking place in birch wood during steaming. When interpreting the differential spectra of cellulose, holocellulose and wood, which according to Nemeth et al. [19], were normalised to the maximum band at 1373 cm$^{-1}$ (i.e., the band characterising stable C—H cellulose binding), we focused mainly on the groups and bonds responsible for the colour change of the steamed wood [20,21] (note: for better visual orientation, the different spectra in Figures 6–8 are shown at the same scale).

The differential spectra of Seifert's cellulose, which was isolated from both the original and steamed birch wood, showed both positive and negative absorbance changes in the 3200–3500 cm$^{-1}$ region (intra-molecular hydrogen-bonded cellulose region) and small band changes at 2800–2900 cm$^{-1}$ (region of C—H vibrations in CH$_2$ groups). These changes are related to changes in the amorphous portion of cellulose. The decrease in absorbance at 1726 cm$^{-1}$ (region of unconjugated carbonyl, carboxyl and ester groups) and the increase at 1639 cm$^{-1}$ (region of conjugated carbonyl) in mode III at 135 °C can be explained by the oxidation reactions of tannin-type extractants and cellulose to form coloured quinone structures, which was also confirmed by the pinkish shade of the prepared Seifert's cellulose.

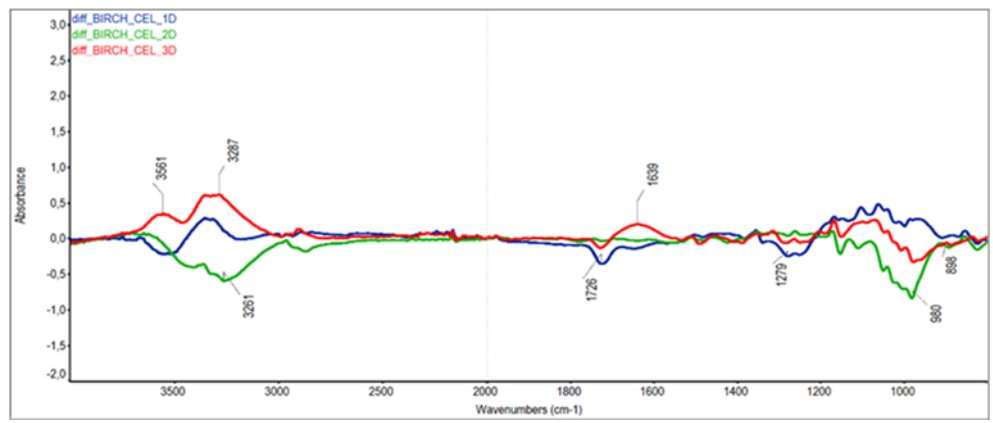

**Figure 6.** The absorption difference spectra of Seifert's cellulose.

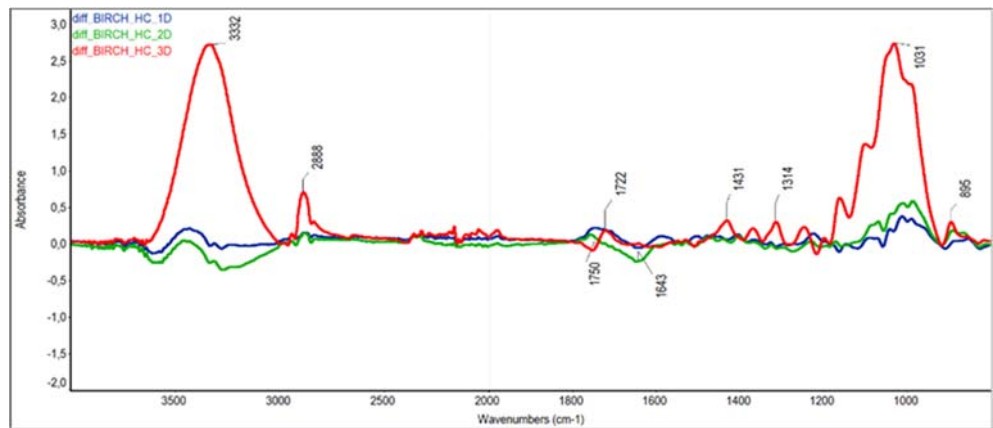

**Figure 7.** The absorption difference spectra of holocellulose.

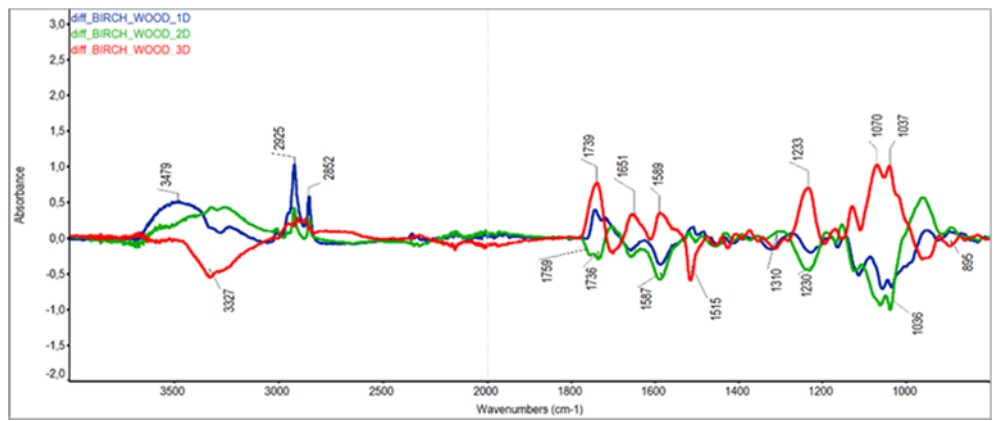

**Figure 8.** The absorption difference spectra of birch wood.

The absorbance of C=O and C—O—C bonds between 950 and 1200 cm$^{-1}$ was generally difficult to interpret due to overlapping of the strong positive and negative absorption bands. There were no significant changes in the cellulose regions studied [19,22].

Holocellulose differential spectra show several changes compared to cellulose differential spectra, some of which can be influenced by the holocellulose preparation method. In particular, a pronounced positive band at 3332 cm$^{-1}$ characterises intramolecular hydrogen bonds in cellulose. The increase in intramolecular hydrogen bonding of the —OH groups observed in mode III was probably due to the degradation reactions of hemicellulose that occurred during the holocellulose preparation. In particular, the positive band of the CH$_2$ group at 2888 cm$^{-1}$ and the band at 1431 cm$^{-1}$ indicated the dehydration reactions of hemicelluloses associated with xylan formation. The changes in absorbance at 1750 cm$^{-1}$ and 1722 cm$^{-1}$ indicated a slight increase in the content of unconjugated carbonyls associated with the cleavage of acetyls from hemicellulose and the formation of acetic acid, as well as the thermal oxidation of labile polysaccharides. A decrease in absorbance at 1643 cm$^{-1}$ in mode II indicated a loss of conjugated carbonyls. From the changes in the absorbance areas of C=O and C—O—C bonds between 950 and 1200 cm$^{-1}$, the overall increase of these groups in the hemicellulose proportion, probably due to thermal oxidation reactions, was inferred as a result of the holocellulose preparation.

In the differential spectra obtained from the birch wood samples, a greater number of changes occurred in the wood under the individual steaming regimes. Unlike the spectra of Seifert's cellulose and holocellulose, the difference spectra of birch wood were not burdened by changes associated with the sample preparation treatments.

Positive bands in modes I (105 °C) and II (125 °C) at 3200 to 3500 cm$^{-1}$, and negative bands at 3327 cm$^{-1}$ in mode III (135 °C), suggest changes in the carbohydrate components of wood—mainly from

the significant degradation of hemicelluloses and the subsequent dehydration reactions. The decrease in absorption bands at 2925 and 2852 $cm^{-1}$ is mainly attributed to the birch wood extractives, whose contents gradually decreased with the steaming temperature. The content of unconjugated carbonyl, carboxyl and ester groups (above 1700 $cm^{-1}$) and conjugated carbonyls (below 1700 $cm^{-1}$) also showed an interesting course. In modes I and II, there was an increase in unconjugated carbonyls, while in mode III, an increase in both unconjugated and conjugated carbonyls was observed. The increase in conjugated C=O groups and red colour is presumably due to hydrolysis and oxidative transformation of polyphenols to dark colour polymers [23].

A negative band at 1515 $cm^{-1}$ confirmed the greater sensitivity of the guaiacyl lignin of broadleaves to elevated steaming temperatures. The absorption region at 1230 $cm^{-1}$ (characteristic of C-O bonds in carboxyls) confirmed the course of thermal oxidation reactions and the formation of new carboxyls in steaming mode III.

Steaming wood using saturated steam, according to Hill [24], should protect the wood from thermal oxidation. However, during the experiment, the steaming was interrupted due to sampling the wood after 3, 6, 9 and 12 h, resulting in repeatedly restoring the oxygen atmosphere in the autoclave.

The observed decrease in the absorbance ratios $H_{1732/2900}$ and $H_{1243/2900}$ demonstrated the cleavage of the acyl (acetyl and formyl) groups from the birch wood hemicelluloses (Table 2). This finding is in accordance with the results of the chemical analyses.

**Table 2.** Absorbance ratios $H_{1732/2900}$ and $H_{1243/2900}$.

|  | $H_{1732/1033}$ | $H_{1243/1033}$ | $H_{1732/2900}$ | $H_{1243/2900}$ |
|---|---|---|---|---|
| Original wood | 0.4055 | 0.3001 | 1.7554 | 1.2989 |
| I | 0.4133 | 0.2995 | 1.7390 | 1.2582 |
| II | 0.3854 | 0.2902 | 1.6375 | 1.2331 |
| III | 0.2958 | 0.2347 | 1.2704 | 1.0080 |

Figures 9–11 illustrate the course of the colour changes of the wood expressed as the colour coordinates L*, a* and b* and the change in the total colour difference ΔE* depending on the content of hemicelluloses and extractive substances.

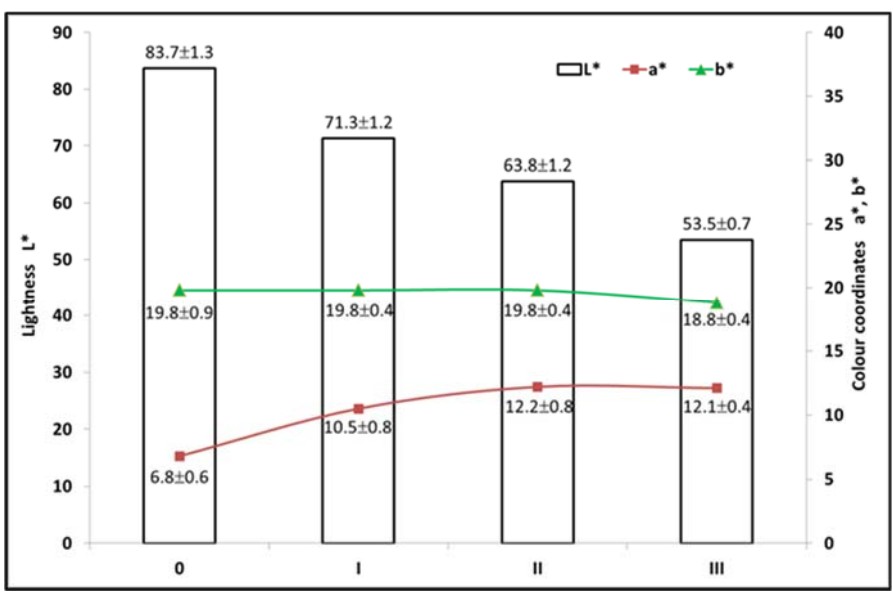

**Figure 9.** Colour coordinates L*, a* and b* according to steaming condition.

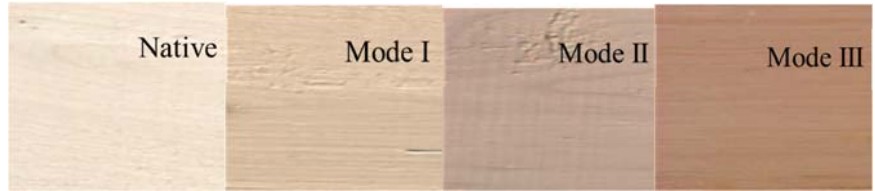

**Figure 10.** View of birch wood before and after heat treatment by individual modes.

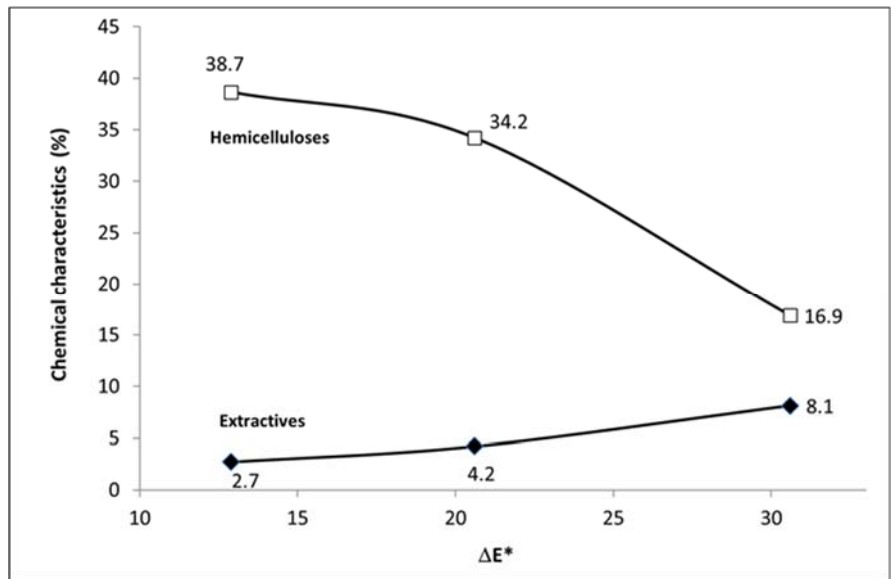

**Figure 11.** Dependence of ΔE* on the content of hemicelluloses and extractive substances.

The lightness L* decreased steadily with increasing steam temperature due to the dehydration reactions of hemicelluloses and formation of secondary coloured substances (furfural and its condensation products). The value of the blue-yellow coordinate b* did not change under mild steaming conditions and decreased from 19.8 to 18.8 in steaming in mode III. In our case, the original extractive substances from wood were probably lost via devolitilisation or thermal degradation. The value of the green-red coordinate a* increased from the original value of 6.8 to 12.2 in steaming mode II and was virtually unchanged in mode III (12.1). The change in a* can be attributed to the air oxidation of water-insoluble polyaromatic structures [23].

To evaluate the colour changes of wood, the total colour difference ΔE* is often used, which involves changes of all three colour coordinates (L*, a*, b*). Figure 10 shows the strong dependence of the total colour difference ΔE* on the content of hemicelluloses and extractives. In particular, the extractives may play a significant role in the formation of secondary colour compounds during hydrothermal treatment. According to previous studies, colour changes of wood can be due to the formation of new chromophore groups that arise from polysaccharide degradation, dehydration products, free radicals and the phenolic hydroxyl groups of lignin [23,25].

Based on a comparison of changes in the colour coordinates L*, a* and b* and selected chemical characteristics of birch wood (hemicelluloses and extractives), it can be concluded that mode II, which does not significantly decrease the hemicellulose content, seems to be the most advantageous.

## 4. Conclusions

The results obtained in this study showed that the change in colour of the wood depends on the steaming conditions and is closely related to changes in its chemical characteristics. The greatest changes in birch wood steaming using saturated steam under operating conditions were observed in

mode III at 135 °C, which resulted in a significant decrease in polysaccharide content, a slight increase in lignin content and an increase in extractive content.

The marked pH shift to the acidic region was accompanied by greater degradation of hemicelluloses and an associated increased production of extractives. The excellent correlation between the pH, the content of extractable substances and total colour difference makes it possible to use the pH to optimize steaming conditions and control the colour modification process for needs of woodworking industry, for example, the manufacture of flooring, wall coverings, furniture made of solid wood, toys and other decorative materials in the home. The dependence of the total colour difference on the pH of wood is a suitable tool for evaluating the achieved colour shade before further technological processing.

A comparison of ATR-FTIR differential spectra in our study has proven to be a suitable method for identifying and studying the causes of wood colour changes caused by steaming. Only minimal changes in the amorphous portion of cellulose were found on the cellulose differential spectra. Several changes in the polysaccharide fraction were found in the differential spectra of holocellulose, but these were largely influenced by the Wise method of holocellulose determination that was applied.

The differential spectra of the birch wood samples indicated that the hemicelluloses were significantly degraded and that the dehydration reactions were able to proceed and to form the dark condensation products. The contents of unconjugated carbonyl, carboxyl, and ester groups increased, but a clear increase in both unconjugated and conjugated carbonyl groups was apparent only in mode III. The findings also confirmed the greater sensitivity of the guaiacyl lignin in broadleaves to elevated steaming temperatures, as well as the course of thermal oxidation reactions and the formation of new carboxyls in steaming mode III.

**Author Contributions:** A.G. and J.G. designed the whole study; J.G., E.V. and M.D. conducted data collection, modeling and results analysis; A.G. and J.G. wrote the original draft paper; A.G. revised and edited the paper. All authors have read and agreed to the published version of the manuscript.

**Funding:** This research was funded by project APVV-17-0456 "Thermal modification of wood with water vapor for purposeful and stable change of wood colour".

**Acknowledgments:** This experimental research was carried out under the grant project APVV-17-0456 "Thermal modification of wood with water vapor for purposeful and stable change of wood colour".

**Conflicts of Interest:** The authors declare no conflict of interest.

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
