# Peer review of "Impact of Steaming Mode on Chemical Characteristics and Colour of Birch Wood"

_forests, doi:10.3390/f11040478_

Round 1

Reviewer 1 Report

The authors do not have sufficiently described the impact of their work in conclusions. What added in conclusions appears as research hints and not as expected impact.

Industry: in which way the presented results can be helpful for industry? and for which kind of industry? The impact is not intended as "future research" but as the benefits that the work can bring.

Similarly, for academia: in which way the presented results and/or the methodology can be considered useful for scholars? Which are the actual benefits that make this work an useful contribution for the scientifi community?

This is not to rise doubts about the work of the authors, but only to push them to better reflect about that an to clearly highlight the actual potentialities of their work. It is deemed useful mainly for authors, since can lead peers to consider their work.

Reviewer 2 Report

The manuscript has been improved in several areas, but it requires further work. Below are the comments:

In the material description there is still no mention for the specific gravity of the birch.

The methods section is still incomplete. For example, in the ethanol-toluene solubility of the wood section you should mention how did you perform the test. Did you do use a separation column, what amount of material and ethanol did you use?, what equipment did you use for the experiment?, etc. Especially with ASTM standards, remember that not all readers will have access to them, and the way and conditions that are used for this specific experiment can vary (brand of the ethanol, unique characteristics of the equipment used, etc.), if you are not specific on your methods, there is no way to repeat this experiment. The same goes for all of the other methods described.

The color reading part has been improved, but the formula is missing and there is still no explanation why the CIE2000 formula was not used (specially considering that you are dealing with a material that has a form factor, compared with a clear liquid). And ISO standard is mentioned, but again, more detail is needed.

Parts of the conclusion can be included as part of the discussion instead. 

The big-picture is still missing. Why is it important to evaluate the chemical changes? How this work makes an impact? 

Round 2

Reviewer 2 Report

The authors have made improvements in the newer version, but some observations are still not covered. Comments below:

Seems that there is a confusion about density and specific gravity. For example, yellow birch has a range of specific gravity from 0.55 to 0.66 (specific gravity does not have units) at a 12%MC. Please read the attached link regarding SG and density: https://www.fpl.fs.fed.us/documnts/fplgtr/fplgtr76.pdf, if you need more references I would recommend reading any wood science book. If, as the authors say, the specific gravity is 704 +- 61, you would be dealing with the most hard wood in the planet (currently this record corresponds to South African black ironwood with an SG=1.49).

In the methods section it would be good to describe how the sawdust for the extractions was obtained (did you use a Wiley mill?, a handsaw?). 

For the ethanol-toluene method, at what temperature was the mixture exposed for the distillation? how many ml of ethanol-toluene was used? for how much time were the samples in the oven?. These sames questions are applicable for the rest of the methods.

In the color reading section, I recommend the authors to add a note indicating the reasoning for the use of the CIE76 formula and not a more recent one, like CIE2000 that is more applicable for color reading of non-uniform materials, like wood.

In the introduction and the results and discussion sections, there is still a lack of the bigger goal of the research. The objective of the research is clear, but why is it important to evaluate the changes of wood under three different steam temperatures?. This overall goal needs to tie with the discussion, as that part is missing on the paper. 

This manuscript is a resubmission of an earlier submission. The following is a list of the peer review reports and author responses from that submission.

Round 1

Reviewer 1 Report

The paper and experimental design is good, but it requires further improvement. Comments below:

It would be helpful to include in the introduction a brief mention to other heat-based treatments, as well as the overall goal of the study (is this a novel method for heat treating wood?)

Line 83 – 86. The scientific name of birch is not mentioned in the methods nor in the abstract. Please add the name including authority. On the same line, there is no information about the specific gravity of the wood, as well as if the sample included sapwood, hardwood or a mix of both. Information about the source of the wood is also missing.

An explanation of why green wood (MC of 45%?), rather than the use of a dry specimen should also be added.

Lines 95-105. The methods stated in that section should be detailed, mentioning the standards only is not enough.

Lines 121-132. What equipment was used for the color reads, as well as which color values were used (SCI or SCE?). Also, considering that wood has a form factor as well as unevenness in the color, why the CIE84 formula was used and not the CIE00? Besides this, how big was the area that the color reader took (30mm, 8mm, 3mm) to determine the L*a*b* values?

Line 191. Scientific writing is in the third person, please correct the sentence

Results and Discussion section. Although the results are interesting the discussion has been done poorly. There are several heat treatments currently used in lab scale as well as industrial scale, how does this method compare to them? Is there a greater hemicellulose degradation at the presence of steam compared to heat? Does a VTC treatment can be simplified with the use of just steam? What is the goal of performing this experiment (the big picture)? None of that is discussed in this section, and although the data gathering is good, it can be improved.

In this same section it would be better to change figure 9 and just have an image of the samples for the readers to observe the color difference of the samples. The independent L*a*b* color comparison is interesting, but it would gain more if the coloration of each compound described is mentioned. As the L* corresponds to the range of lightness, it is obvious that if there is degradation of polysaccharides within the wood, the resulting colors will shift towards a darker spectrum. A similar color comparison should be done for a* (red to green) and b*(yellow to blue), without a direct comparison of the color of the wood compounds (eg. what is the original color of the polyaromatic structures and what is the expected color shift under oxidizing conditions?) this section lacks value.

The study would also gain a lot of importance if for example, the steam treatment allows the wood to resist decay without the need of a preservative or if it improves the dimensional stability of the material. This is a topic that might be interesting for a future follow up of this research.

Reviewer 2 Report

The paper describes an investigation aimed at evaluating the changes in birch wood chemical components caused by steaming at three temperatures – 105 °C, 125 °C and 135 °C. Authors observed the most significant changes in characteristics at 135°C. Additionally, they observed that the hemicelluloses were significantly degraded and that the dehydration reactions were able to proceed.

The paper is well written and the scientific method is described in a comprehensive way. Results are also well presented.

However, the actual impact of the work is not clear. Authors are invited to provide a thorough description of the impact that the observed results can have on both research and industry/forestry.